# Feasibility and Acceptability of a School-Based Emotion Regulation Prevention Intervention (READY-Nepal) for Secondary School Students in Post-Earthquake Nepal

**DOI:** 10.3390/ijerph192114497

**Published:** 2022-11-04

**Authors:** Megan K. Ramaiya, Caitlin L. McLean, Manjila Pokharel, Kiran Thapa, M. Andi Schmidt, Martha Berg, Jane M. Simoni, Deepa Rao, Brandon A. Kohrt

**Affiliations:** 1Department of Psychiatry and Behavioral Sciences, University of California, San Francisco, CA 94143, USA; 2VA San Diego Healthcare System, University of California, San Diego, CA 92161, USA; 3Transcultural Psychosocial Organization Nepal, Kathmandu 44600, Nepal; 4College of Public Health, University of Georgia, Athens, GA 30602, USA; 5School of Graduate Psychology, Pacific University Oregon, Forest Grove, OR 97116, USA; 6Department of Psychology, University of Michigan-Ann Arbor, Ann Arbor, MI 48103, USA; 7Department of Psychology, University of Washington, Seattle, WA 98195, USA; 8Department of Global Health, University of Washington, Seattle, WA 98104, USA; 9Division of Global Mental Health, George Washington University, Washington, DC 20052, USA

**Keywords:** non-randomized controlled trial, emotion regulation, dialectical behavior therapy, schools, adolescents, Nepal, global mental health

## Abstract

Background: Child and adolescent mental health problems are major contributors to the global burden of disease in low- and middle-income country (LMIC) settings. To advance the evidence base for adolescent mental health interventions in LMICs, we evaluated the feasibility and acceptability of a school-based emotion regulation prevention intervention (READY-Nepal) for adolescents who had a recent exposure to a humanitarian disaster. Methods: A mixed-method, non-randomized controlled trial was conducted with Nepali secondary school students in one heavily affected post-earthquake district. Students (*N* = 102; aged 13 to 17 years) were enrolled in the intervention (*n* = 42) and waitlist control (*n* = 60) conditions. Feasibility and acceptability were examined via attendance, and by qualitative interviews with a subset of students (*n* = 15), teachers (*n* = 2), and caregivers (*n* = 3). Preliminary efficacy was examined on primary outcome (emotion regulation) and secondary outcomes (anxiety symptoms, posttraumatic stress symptoms, functional impairment, resilience, coping skills), which were measured at baseline and post-intervention (four weeks). Results: Delivering the intervention was feasible and acceptable, as demonstrated by low dropout (8%) and high program attendance (6.7 of 8 sessions). Qualitative data suggested high uptake of anger regulation skills, but lower uptake of mindfulness skills. Despite this, there were no significant differences by condition on primary or secondary outcomes at four-week follow-up. Students provided suggestions for improvement of the program. Conclusion: Further research on longitudinal outcome measurement, use of alternatives to retrospective self-report data, and rigorous development of culturally grounded models of emotion regulation is necessary to explore the utility of school-based emotion regulation interventions in Nepal and other LMICs.

## 1. Introduction

Child and adolescent mental health problems are major contributors to the global burden of disease [1]. Mental health problems affect an estimated 10–20% of adolescents, with youth in low- and middle-income countries (LMICs) disproportionately impacted due to high levels of political unrest, economic and social hardship, and chronically low access to evidence-based mental health services [2]. The World Health Organization (WHO) estimates that 14% of adolescents (age 10–19 years) worldwide live with a current mental disorder, with half of mental disorders in adulthood developing by the age of 14 years [3].

Though factors contributing to mental health symptomatology among youth are heterogeneous, exposure to environmental and political trauma has emerged remains a consistent risk factor underlying the development of mental health and psychosocial problems [4]. Prior studies in LMICs have demonstrated a higher prevalence of psychiatric and psychological disorders among children and adolescents exposed to political violence [5], as well as among youth exposed to a wide range of natural disasters [6]. In Nepal, specifically, a violent 7.8 magnitude earthquake struck Nepal’s central region in 2015, resulting in an estimated 9000 deaths and 22,000 injuries as well as significant damage to homes, government buildings, health facilities, and road networks [7]. Post-earthquake estimates indicated a rise in depression, anxiety, and stress-related symptoms, with higher needs of support services indicated for women and adolescents under the age of 18 [8,9,10]. Although a host of foreign and non-governmental services were rapidly deployed in affected districts during the immediate aftermath of the disaster, services were often brief, isolated, and fragmentary, with virtually no programs resulting in coordinated efforts to sustainably address a large number of affected individuals.

School settings provide a unique opportunity for addressing the mental health needs of conflict-affected youth. Schools are found in abundance throughout both the urban and rural landscape in LMICs, with overall rates of attendance dramatically improving since 2000 [2]. In a number of high-income countries (HICs), they have provided a scalable and efficient platform to target risk factors and strengthen protective factors prior to peak emergence of mental health conditions. They also provide a natural and readily accessible source of trainable manpower in the form of teaching professionals, which may be a critical component in reducing the treatment gap in LMICs. Opportunities for integration within the broader education system also make school-based settings attractive locations for sustainably addressing youth mental health needs.

The rising interest in these programs mirrors recognition of the mental health needs of school-going adolescents globally. Due to economic and social pressures at individual and family levels, for instance, many school-going adolescents in LMICs are tasked with adult responsibilities at a young age, with added social and financial obligations compounding mental health stressors commonly experienced during this developmental period [11]. In a study of adolescent student disaster survivors in Nepal, 26.8% and 50.3% had probable post-traumatic stress disorder (PTSD) and depression, respectively [12]. Daily exposure to high levels of household and interpersonal stress was strongly associated with PTSD, depressive, and anxiety symptoms (OR = 1.49−1.90), above and beyond exposure to disaster trauma. Another study reported 11–40% and 23–40% of adolescents experienced post-traumatic stress and depressive symptoms, respectively [10]. Despite the sizeable psychological burden present in this demographic, however, mental health provision has been chronically unprioritized in LMICs and further compounded by issues ranging from community and provider stigma, poor training, wide implementation of non-evidence-based interventions, and minimal integration into existing healthcare infrastructures.

One barrier to addressing youth mental health needs in LMICs is the weak and mixed evidence base for the efficacy of school-based programs. To date, only seven rigorous trials have been examined the effects of psychosocial interventions designed specifically for LMIC, trauma-exposed youth [2]. In Indonesia, one trial demonstrated efficacy of a classroom-based intervention in reducing post-traumatic stress (PTS) symptoms among school children [13]. Other trials in Sri Lanka [14] and Palestine [15] implementing a similar intervention, however, found no detectable effects on mental health outcomes between experimental and control conditions. In one systematic review of these and other school-based treatment studies for conflict-affected youth, results indicated a lack of rigorous studies, use of divergent interventions, a skewed PTS focus, and mixed results, with less than half of the 22 studies resulting in positive findings and a fraction supporting iatrogenesis [2]. In Nepal, specifically, both study designs and results of classroom based interventions have been heterogeneous. In one randomized evaluation of a classroom-based intervention for conflicted-affected rural students with moderate-to-severe distress [16], there were gender-specific improvements in socio-behavioral symptoms, but no psychiatric symptom improvements. In a subsequent randomized controlled study targeting primary prevention among earthquake-exposed students through teacher-led psychosocial support provision [17]. there were no improvements in PTSD or depressive symptoms. And, in one non-controlled study of school-based Interpersonal Therapy delivered by non-specialist providers, students with depressive symptoms and functional impairment experienced improvements across both categories [18]. Given this scarce and heterogeneous evidence base for classroom-based interventions in Nepal and other resource-strained settings, it is therefore crucial to implement novel treatment outcome studies.

## 2. Objective and Methods

### 2.1. Objective

The objective of this study was to contribute to the narrow evidence-base for psychosocial interventions for adolescents exposed to environmental and disaster-related trauma in LMICs. This paper describes a pilot trial to evaluate the preliminary efficacy of a novel brief, school-based emotion regulation intervention targeting broad prevention of mental health symptomatology and increases in positive mental health among disaster-exposed youth in Nepal.

We chose dialectical behavior therapy (DBT; [19,20]) as the overarching treatment model for this population for a number of reasons. First, DBT is a cognitive behavioral therapy that emphasizes a flexible, contextual, and principle-driven view of behaviors [21]. Delivery of DBT requires tailoring of techniques or strategies to the client’s unique set of circumstances. Because of this inherent flexibility, it was considered ideal for cultural modification with ethnic Nepalis. Second, DBT’s emphasis on teaching practical, real-world skills via group setting allowed for development of a manualized protocol for use by lay Nepali counselors or other paraprofessionals with limited-to-no prior mental health experience. Third, DBT explicitly integrates Zen Buddhist principles, mindfulness, and acceptance into treatment [22]. Its conceptualization of clients and events using these Buddhist perspectives had the theoretical potential to align with Nepali ethnopsychological divisions of the mind, body, and self [23]. Ethnopsychology can be understood as the study of cultural or “folk” models of psychological subjectivity [23] and is considered a powerful means of uncovering a culture’s own understanding and experience of the self, emotions, physical body, and connections to the social world. Theory on Nepali ethnopsychology was generated from two decades of ethnographic research with Nepali-speaking populations, including participant observation and participant interviews with traditional healers, biomedical medical practitioners (e.g., psychiatrists and general physicians), and psychologists. Research on Nepali ethnopsychology has been used to guide theoretical [24] and empirical [25,26] adaptation of DBT, as well as other evidence-based psychosocial interventions in Nepal [27]. Fourth, DBT’s emphasis on treating core processes of emotion dysregulation [28] allowed us to target a wide and flexible range of transdiagnostic mental health problems common among adolescents more broadly. (See Figure 1 for a description and explanation of an adapted Nepali variant of the emotion regulation model.) Prior research in Nepal has demonstrated the promising utility of emotion regulation-focused interventions in reducing a range of psychopathology (e.g., depression, anxiety, posttraumatic stress) in both youth and adult populations [25,26]. Lastly, Nepali service providers had expressed interest in a DBT-informed intervention, with members of the larger community also indicating they would be receptive to treatments drawing from various Eastern cultural traditions. Combined, these factors recommended development and pilot evaluation of a DBT-informed intervention.

This theoretical model synthesizes data on processes generating and maintaining emotion dysregulation developed in HIC [19] with Nepali ethnopsychological adaptations. According to Nepali ethnopsychology, the physical body is divided into a number of components, including the heart-mind (Nepali *man*, the location of affect, memory, and uniqueness) and brain-mind (Nepali *dimaag,* the rational that simultaneously governs normative social behavior). According to the model, external or internal events are thought to trigger emotional heaviness (e.g., stress) in the *man*. Heart-mind problems result in socially inappropriate behaviors, which are then socially invalidated. This feedback loop, when coupled with an early predisposition (genetic, environmental and developmental) for emotional sensitivity and reactivity, results in a transacting model of pervasive dysregulation. If left untreated, more severe externalizing psychopathology (e.g., suicide attempts, substance use disorders) are theorized to result.

The resulting intervention, Regulating Emotions through Adapted Dialectical Behavior Skills for Youth (READY-Nepal), is a manualized, emotion-focused skills training program designed to promote positive mental health and support resilient responding in trauma-exposed adolescents. (See Table 1 for an overview of the curriculum, and Figure 2 for sample Nepali handouts adapted for school-going youth.) The program consists of 8 group sessions, each lasting approximately 50 min in length. It was provided twice-weekly for a total of four weeks, although alternative delivery formats are also possible. The modularized, emotion-focused intervention was informed by principles of DBT and was designed to both augment and generalize to Nepali adolescents’ emotion regulation abilities during stressful experiences. The intervention was developed from a prior, more intensive version of the intervention that was culturally adapted and piloted using community-based participatory methodology with a cohort of low-literacy women with histories of suicidal behaviors in Nepal [25,26]. Session length and content were refined using co-development workshops with TPO-Nepal (the collaborating research organization) and school administrative staff (Ramaiya et al., unpublished). The program is divided into 5 components, and includes both didactic and experiential instruction in skills related to mindfulness, stress tolerance, emotional awareness and regulation, and mastery of interpersonal relationships. All sessions begin with a group mindfulness exercise, followed by a behavioral chain analysis to identify target problem behaviors appropriate for skills training. Skill-specific didactics and experiential activities for one of the three READY-Nepal modules (Mindfulness, Emotion Regulation, and Interpersonal Effectiveness) followed. Each session concluded with homework assignment and generalization strategies for skills learned during the session. The intervention was delivered in Nepali by two local research assistants with no prior mental health experience and a US doctoral student with one year of comprehensive DBT training. An experienced, US-based DBT clinician provided regular supervision. The intervention was delivered at no financial cost to participants.

The handout on the left is an adaptation of DBT Wise Mind. The left portion of the handout refers to the brain-mind *(dimaag)*, with associated images. The right side represents the heart-mind *(man)* division of self. The overlapping portion in the diagram signifies wise mind *(buddhi mani soch)*. The handout on the right is an adaptation of DBT Observing & Describing Emotions (Shame). The text above the roots signifies emotion components visible to others, while that below the roots signifies private emotion components.

Our primary research question concerned the feasibility, acceptability, and preliminary efficacy of READY-Nepal delivered in classroom settings. We aimed to examine if secondary school students self-selected into the intervention, the number of sessions attended by students, and that the measures were able to be completed. Additionally, we preliminary compared changes on a broad range of outcome measures relative to a wait-list control condition. Based on prior initial findings of gender-based differences in group interventions in Nepal and other LMIC [18,30,31] our second research question concerned differential intervention effects based on gender. We examined if females, relative to males, would be more engaged in the intervention and experience greater improvements on outcome measures.

All portions of the study were conducted in collaboration with Transcultural Psychosocial Organization (TPO)-Nepal, a mental health research-oriented organization based in Kathmandu. The study received approval by the Nepal Health Research Council and Duke University’s Institutional Review Board (Pro#00071881).

### 2.2. Methods

#### 2.2.1. Setting

Nepal has a population of approximately 30 million [32], of which 17.4% are multidimensionally poor [33]. Nepal suffered a 10-year long civil war between the Communist Party of Nepal and government forces leading to deterioration of fragile socioeconomic and mental health infrastructures, which have since seen minimal lasting improvements. Government mental health resources are scarce, with virtually nonexistent access to formal mental healthcare in rural as well as many urban and peri-urban settings [34]. The overall ratio of psychological resources per capita is negligible, with 0.12 psychologists available per 100,000 persons [3].

The specific study was conducted in Sankhu, a small population center in central Nepal that comprises approximately 2400 people [33]. On account of its location in the primary earthquake epicenter region in Kathmandu valley, the effects of the 2015 earthquakes were pronounced in the region. In Sankhu town center, located in close proximity to the second earthquake’s epicenter, 45 people were killed and 1200 homes destroyed. Schools were closed for approximately two months after the earthquake, and rebuilding remains an ongoing project. The present study took place from June to September 2016 (14 months after earthquake exposure).

#### 2.2.2. Design

A mixed-method, gender-stratified, non-randomized controlled design was conducted targeting Nepali secondary school students. Classrooms were not randomly assigned to conditions due to uneven school requirements. Classrooms with limited time available to participate in the intervention were assigned to the waitlist control condition, while those with availability were assigned to the intervention condition. In order to reduce the potential for contamination effects among students (i.e., diffusion of skills or related intervention content), participant groups were defined to match the school’s existing classroom designations, further subdivided only by gender. Each resulting group was then assigned as a whole to either experimental or control conditions. Quantitative outcome measures were taken on two occasions: one-week pre-intervention and four weeks post-intervention.

#### 2.2.3. Quantitative Outcome Measures

Quantitative outcome measures were used to evaluate changes on a range of indicators, ranging from emotional and psychological difficulties, positive aspects of well-being, and mental health symptomatology. All quantitative data was collected pre-intervention and 4-week follow-up via a self-administered, tablet-based battery. This method was preferred over typical verbal administration due to the literacy level of participants, in addition to the need to maximize potential reductions in available time for participating students.

#### 2.2.4. Primary Outcome

**Emotion Regulation.** Changes in emotion regulation were assessed using an adapted version of the Difficulties in Emotion Regulation Scale (DERS; [35]). The DERS, a 36-item self-report scale, is designed to assess multiple domains of emotion regulation. In Nepal, the instrument has been adapted for the sample using a validated transcultural translation process [36]. Each item is scored on a four-point scale from 1 “almost never” to 5 “almost always,” with higher scores indicating greater of emotion dysregulation. Examples of backtranslated items from the Nepali version include “In the last two weeks, when I was in emotional pain, I didn’t know if I was sad, frightened, or angry” and “In the last two weeks, I knew exactly what feelings were happening in my heart-mind”. Internal consistency in this study was α = 0.77.

#### 2.2.5. Secondary Outcomes

**Coping Skills.** Uptake of coping skills was assessed using a variant of the Dialectical Behavior Therapy Ways of Coping Checklist (DBT-WCCL): The DBT-WCCL is an adaptation of the Revised Ways of Coping Checklist (RWCCL; [37]) that includes additional items intended to represent DBT-specific skills [38]. In Nepal, the instrument has been transculturally adapted and tested with a sample of suicidal and self-harming women in a rural district. The 18-item Nepali adapted version measures the frequency of DBT skills use over the previous two weeks. Examples of backtranslated items include “In the last two weeks, I took care of my body and so that I was less emotionally sensitive (e.g., got a good night’s sleep, ate properly, avoided mood-altering drugs)” and “In the last two weeks, I did puja or went to temple”. The Crohnbach’s alpha for the DBT-WCCL in this study was 0.73.

**Anxiety.** Changes in prior to-week adolescent anxiety symptoms were assessed using the Nepali version of the Beck Anxiety Inventory (BAI; [39]). The BAI, a 21-item instrument, has been clinically and culturally validated use in Nepal. Items are scored 0–3 with an instrument rage of 0 to 62. Based on clinical validation in Nepal, a score of 17 or higher indicates moderate anxiety with need for intervention. Examples of backtranslated items include “dizziness” and “fear of dying”. Crohnbach’s alpha for the 21-item BAI was 0.90. A subset of six items from the original BAI were used in this study, based on items most frequently endorsed in a longitudinal study examining the mental health among Nepali adults affected by conflict [40].

**Child Post-Traumatic Stress.** PTS symptoms in the prior two weeks were measured using the child version of the Posttraumatic Symptom Scale (CPSS; [41]). The 17-item self-report measure correlates with posttraumatic stress disorder diagnostic criteria outlined in the *Diagnostic and Statistical Manual of Mental Disorders* (DSM-IV; [42]). The sensitivity in Nepal is 0.68 and specificity is 0.73 at a score ≥ 20 (Kohrt et al., 2011). For this study, we used five items demonstrating high discriminant validity: nightmares (CPSS 2), flashbacks (CPSS 3), traumatic amnesia (CPSS 8), feelings of a foreshortened future (CPSS 12) and easily irritated at small matters (CPSS 14). Crohnbach’s alpha for the CPSS was 0.81.

**Functional Impairment.** Changes in two-week adolescent functional impairment were measured using the Nepali version of the Child Functioning Impairment Scale (CFI; [43]). The CFI is a 10-item scale assessing functional impairment in a number of domains (difficulties working in the field, completing household chores and schoolwork, bathing), with higher scores indicating greater functional difficulties. The CFI has been clinically and culturally validated for use in Nepal, with a Crohnbach’s alpha for the Nepali version of 0.78.

**Resilience.** Changes in two-week adolescent resilience were measured using items adapted from the Wagnild & Young Resilience Scale [44]. This adapted scale, 7-item has been adapted for use in Nepal and shows a strong association with biological markers of resilience [45]. Items assess levels of resilient behaviors across personal and social categories, and are scored from 0 “never” to 3 “every time” with an instrument rage of 0 to 62. Examples of backtranslated items include “In the last two weeks, how often did you feel like you could depend on others?” and “In the last two weeks, how often did you feel you could feel happy, even when you are sad”? All items demonstrate an association with depression and anxiety, and have an item-total correlation of greater than 0.4. Internal consistency of the total score was 0.77 in the Nepali version.

**Suicidal Ideation**. Changes in past-two-week suicidal ideation, was assessed via a 4-item scale with “yes” and “no” response type options. Students with answers of “yes” were assessed for risk by a study team member (MKR; a clinical psychology doctoral student with prior training in Nepali suicide prevention). Those assessed as having high risk (i.e., planning, access to means, and intent to die) were referred to an on-site counselor at TPO-Nepal for risk management.

### 2.3. Qualitative Measurement

Three semi-structured interview guides were separately developed and completed at four-week follow-up with students in both conditions (*n* = 10), teachers (*n* = 2), and caregivers of participating students (*n* = 3) to explore perceived feasibility and acceptability of the intervention. The qualitative sample was recruited via purposive sampling with stratification by gender (female vs. male). Recruitment was based on availability and study interest.

Guides were developed collaboratively with the research team and TPO-Nepal and included the following a priori themes: generalized perceptions of the intervention, perceived benefits, suggested improvements, and skills use. Guides were translated into Nepali by two bilingual research assistants (both native Nepalis) and pilot tested with clinicians at TPO-Nepal, with minor revisions made for clarity and flow.

### 2.4. Data Collection

Three local research assistants and two US masters-level researchers who were not involved in service delivery were selected and received one week of quantitative and qualitative research training. It was not possible to blind assessors to treatment status, as they were required to visit the site to conduct both quantitative assessments and qualitative interviews.

### 2.5. Quantitative Analyses

All statistical analyses were conducted using SPSS Version 28.0 [46]. First, score distributions across all continuous dependent variables were examined for normality, with appropriate transformations made to normalize distributions with excessive skewness (>2.0) or kurtosis (>7.0). Independent-samples *t*-tests were used to compare completers vs. dropouts in addition to intervention vs. control participants. Baseline differences in general characteristics and outcome measures were then assessed via Wilcoxon rank sum test for continuous measures or Chi Square tests for categorical measures. Correlations between baseline outcome variables are presented in Table 2. Conditions were compared using a series of Repeated Measures Analysis of Covariance (ANCOVA). We tested for the inclusion of possible covariates (e.g., gender, age, caste, socioeconomic status) and gender was found to be significantly correlated with outcomes at baseline and retained in the final models. Because of our interest in the differential impact of the intervention based on gender, gender was examined as a moderator using a series of hierarchical regressions on follow-up outcomes, while controlling for pre-intervention outcome levels.

### 2.6. Qualitative Analyses

Semi-structured interviews were used to complement quantitative data by providing preliminary data relevant to the intervention’s broader feasibility, acceptability, and dissemination potential. Interviews were transcribed, translated, and reviewed for accuracy. QSR NVivo software version 11.0 was utilized for data management and analysis. The analytic approach was informed by thematic analysis. All of the English transcripts were carefully reviewed by two members of the research team, with both identifying patterns to inform codebook development through this collaborative review. When the codebook was finalized, 20% of the transcripts were double coded with discrepancies discussed until 90% inter-coder agreement was reached. Themes emerged as comparisons within and across individuals were made, and representative quotations for each of the identified themes were selected

## 3. Results

### 3.1. Description of Participants

Study participants (*N* = 102) were adolescents aged 13 to 17 years, attending one earthquake-affected secondary school in Sankhu identified through key informants at TPO-Nepal. Allocation to study conditions (See Figure 3) followed a two-step procedure using purposive sampling using whole classrooms. First, a pool of eligible classrooms (*n* = 6) was identified through collaboration with school administrators through use of an official class registry. Classrooms were eligible for participation if students were between the ages of 13 and 17. Next, whole classrooms were non-randomly assigned to either intervention or control conditions. Classrooms were eligible for enrollment in the intervention condition if they (1) had a free period during the allocated intervention time; (2) had a relatively equal proportion of males and females; and (3) students were not preparing for a national secondary school examination at the time of intervention delivery. Classrooms enrolled in the intervention condition (*n* = 2) were then segregated by gender, leading to the formation of four groups (*n* = 40 participants). All other classrooms (*n* = 4; 62 participants) were assigned to the wait-list condition, who received the intervention the subsequent year.

Demographic and clinical characteristics and suicidal behaviors are summarized in Table 3. The total sample included an approximately even divide between males (49%) and females (51%). A majority of participants (71.9%) were members of the Newari ethnic group. At baseline, 29.2% of participants (*n* = 28) reported current (i.e., over the prior two weeks) suicidal ideation. At baseline, 14.6% of participants (*n* = 14) reported any lifetime suicidal behavior (i.e., suicide attempts, non-suicidal self-injury episodes).

The qualitative sub-sample of students (*n* = 10) included six females (60%) and four males (40%) distributed between grade 7 (50%), grade 8 (10%), and grade 9 (40%). The mean age was 14 years. A majority were Newaris (60%), one was high-caste Brahmin (10%), one was low-caste Dalit (10%), and two self-identified as other (20%). Teachers (*n* = 2) were 50% female. Caregivers (*n* = 3) were 100% female and included two biological mothers and one grandmother.

### 3.2. Dropout & Session Attendance

Average dropout (defined as missing more than 50% of sessions) was low (8%), with rates for females (6%) lower than males (9%). Overall attendance in the 8-session READY-Nepal program was high (*M* = 6.7), with females attending a higher mean number of sessions (7.1) relative to males (6.3).

### 3.3. Quantitative Clinical Outcomes

Data for coping skills use, anxiety, PTS, and functional impairment were positively skewed. Z scores for these outcome variables improved to acceptable normality parameters for following square root transformations. After transformation, there were between one to four baseline outliers for emotion regulation, functional impairment, and resilience, which were retained in the final analyses. There were comparable percentages of missing student data at baseline and follow-up, and no significant baseline differences between conditions on any outcome variable.

Table 4 presents results of the Repeated Measures ANCOVAs for primary and secondary outcomes. There was a significant main effect of time for functional impairment (*p* = 0.004), such that impairment significantly decreased over time across both conditions. Additionally, there was a significant time*gender interaction for Anxiety (*p* = 0.011), with females reporting lower anxiety symptoms relative to males across both conditions. No other significant differences were found. Hierarchical regressions examining gender as a moderator found no significant gender*condition interaction effects (all *p*s > 0.135). See for Figure 4 for graphs of outcome changes by condition and gender.

The first column represents full-sample changes by condition, while the second column segregates pre- and post-intervention differences by gender. Blue and red lines represent the intervention and control conditions, respectively.

Due to the small base rate of suicidal behaviors by condition, statistical modeling changes in suicidal behavior was not appropriate. Table 5 presents counts of self-reported suicidal ideation in the prior two weeks. Frequency of suicidal ideation decreased by 23% in the intervention condition (*n* = 10 from *n* = 13) and remained unchanged in the control condition. For females, suicidal ideation decreased in intervention condition (*n* = 7 from *n* = 9) and increased in the control condition (*n* = 10 from *n* = 7). For males, suicidal ideation decreased in both the intervention (*n* = 3 from *n* = 4) and control (*n* = 5 from *n* = 8) conditions.

### 3.4. Qualitative Results

Table 6 presents results and illustrative participant quotations from qualitative analyses. A majority (*n* = 7; 87.5%) of adolescents in the READY-Nepal condition preferred a group format with separation by gender, highlighting acceptability of these two program components. All females (*n* = 5) and males (*n* = 3) highlighted specific utility of opposite action to anger. Three indicated that mindfulness skills (“especially wise mind”) were difficult to grasp conceptually, and that skills taught through the curriculum would be better-suited for older youth populations (age 18–21) due to their relatively higher level of abstraction. Participating students also recommended a number of improvements, including decreased session length (from 1 h to 45-min to accommodate classroom schedules) and a greater total session count (e.g., 4–6 additional sessions or follow-up “booster” sessions), inclusion of a reward system to facilitate engagement, inclusion of family members into the intervention, and an increased focus on social interaction skills.

For students in the control condition (*n* = 2), one female shared that females participating in the intervention had shared skills with her (e.g., opposite action for anger), and this was less common for males. Teachers (*n* = 2) expressed a range of opinions on the intervention, with one recommending greater system-level coordination between facilitators, teachers and school administration to support teacher’s ability to reinforce and generalize skills use in the classroom setting. Caregivers (*n* = 3) of students participating in the intervention suggested minimal sharing of content learned at home, and mixed improvements.

## 4. Discussion

The current study used a mixed-method, gender-stratified, controlled design to investigate the feasibility, acceptability, and preliminary efficacy of an 8-session, skill-based emotion regulation curriculum in secondary school students in a post-disaster Nepali setting. Because a vast majority of evidence-based, universal school prevention programs are developed and disseminated in HICs, our study fills an important gap by examining the extent to which these interventions generalize to non-western, LMIC settings with high rates of traumatic exposures and unique contextual barriers and facilitators.

Study findings indicated low dropout (8%) and high mean session attendance (6.7), with retention rates higher for females relative to males. The low dropout rate is parallel with examinations of school-based mindfulness interventions delivered in HIC, and lower than existing trials of school-based mindfulness interventions in Asia (41% dropout; [47]) and DBT skills-specific groups in secondary schools in HIC (15% dropout; [48]). These results suggest feasibility of emotion regulation skill training curricula of READY-Nepal in school environments in post-disaster contexts.

Qualitative findings complemented dropout and attendance data, by indicated feasibility and acceptability of specific program content. A majority of adolescents in the READY-Nepal condition shared a preference for group-based (over individual) sessions to promote social interaction and connection. A majority also preferred group separation by gender, and believed that this allowed for discussion of gender-specific topics and promoted within-gender connectedness. All adolescents recalled and highlighted the utility of opposite action to anger, suggesting that this led to a calm heart-mind and facilitating healthy social relationships.

Despite promising feasibility and acceptability, we found no quantitative improvements in either primary (emotion regulation) or secondary outcome variables (anxiety, posttraumatic stress, functional impairment, resilience, coping skills) at four-week follow-up between study arms. These findings exist against a backdrop of mixed findings on school-based emotion regulation interventions. In one recent meta-analysis of emotion regulation psychosocial interventions for adolescents in community-based and war-affected settings [49]), there were mixed results regarding immediate post-treatment efficacy, with the largest effect sizes observed among youth with clinical diagnoses, and the smallest sizes found in studies implementing prevention programs for community youth. A parallel systematic review on efficacy of DBT skills groups among school-going youth [50] also showed mixed immediate post-treatment results in primary and secondary school students in HIC (vs. replicated efficacy in college student populations).

Given these prior findings as well as our own, one explanation for the lack of significant quantitative findings is the presence of floor effects, which are common in universally delivered programs given the relative health of the sample compared to a clinical group. Of existing empirically supported, school-based interventions in LMICs, half have utilized targeted screening to only enroll select students (e.g., those with demonstrated mental health problems, risk factors for mental illness, and/or current symptoms of PTS; [2]). In contrast, only 38% of LMIC studies adopted a “universal” approach like ours by selecting whole classes, with half of these studies (*n* = 4) reporting mixed or null findings. Further exploratory analyses that aim to identify sub-groups or sub-populations of students that benefitted from the READY-Nepal intervention is thus warranted, in order to identify individuals that may derive maximal benefit from the intervention. It is of note that, given the higher proportion of mental health symptomatology in our sample relative to other similar-age samples [51,52], these floor effects do not singularly explain our findings and justify the need for additional investigation.

Our measurement selection and timing of delivery may also influence quantitative findings. The original DERS was developed to measure dispositional tendencies (vs. state-level emotion regulation processes that can be altered via interventions; [35]) and may therefore mask potential observed effects. The retrospective self-report nature of the scale, as well as all others in the assessment battery, may lead to biased and therefore inaccurate recordings of behaviors. The frequency and timing of measurement (i.e., only baseline and immediate post-treatment measurement) may also explain lack of detected main and secondary effects. Studies of adolescent emotion regulation interventions in HIC indicate mixed findings upon inclusion of follow-up periods [49], with some promising work showing increased gains over time as skills begin to solidify and generalize. Future research should consider more intensive follow-up measurement, in addition to employment of alternative measurement models (e.g., ecological momentary assessment of emotion regulation, observational ratings from program facilitators, observable behavioral outcomes).

An additional competing explanation for quantitative findings is that DBT skills require higher-level abstraction and metacognitive abilities that are developmentally less prevalent among youth (age 12–17) who comprised our sample. Mindfulness skills, which comprise the foundation of DBT skills training curricula, are one example of a metacognitive series of behaviors requiring the individual to not only recognize their own cognitive thinking patterns but also decenter from them. Our qualitative findings showed low uptake of mindfulness skills by students, which has also been show in prior studies with non-Western populations [53]. Qualitative findings also showed increased recall and use of opposite action to anger across both genders, which may be because anger is a high-arousal emotion that is easy to identify and therefore change. On the contrary, Western mindfulness skills, as well as overlapping emotion regulation skills (e.g., emotion awareness, emotion acceptance) may also reinforce and be a product of individualist self-construals (e.g., focusing on independent internal experiences), which may be unfamiliar to South Asian adolescents holding collectivist values [54]. Further clarification of pathways linking relevant cultural dimensions (e.g., individualist and collectivist values related to emotion and construal of the self, somatization, and stigma, among a host of others) to the generation and maintenance of emotion regulation in Nepali adolescents is necessary to understand how school-based interventions may generate meaningful change. Potentially differing sources of traumatic stress in the sample and their relationship to dysregulation are also unknown (e.g., direct earthquake effects, poverty, family relations, community violence, bullying), which precluded targeting of intervention content to salient youth presenting problems. Until these pathways are clarified (and accompanying intervention targets identified), program developers may continue to “fly blind” in their attempts to implement theoretical models with potential limited applicability in LMIC populations.

Questions regarding optimal dosage of school-based interventions in LMIC populations remain to be explored. In one systematic review of school-based mental health programs in LMICs [2], the average length of sessions ranged from eight to 20 sessions, with a mean of 14 sessions. This average is nearly twice the number of sessions included in the current READY-Nepal program, which consisted of a smaller number of sessions (based on feedback from co-development workshops with Nepali researchers and school administration) to enhance feasibility of implementation in a rural, school setting. Although reduced session count has potential to enhance implementation, both qualitative and quantitative findings suggest that a greater dosage may enhance clinical utility of the intervention. Future studies might formally investigate optimal session number and length, or whether alternative strategies to increase the dose of the intervention (e.g., short daily classroom practices, extending curriculum length or teaching additional modules over subsequent year levels) impact findings.

A number *(n* = 6) of trials in LMICs, and an even larger number in HICs, have utilized a health promoting schools model by which socio-emotional learning is embedded into the school’s existing health education curriculum and is taught at the whole-school level [2]. Although the modular READY-Nepal curriculum has the potential to be integrated using a similar model, the current trial was delivered as an adjunctive intervention. Further, most health promoting schools intervention content relies on use of cognitive behavioral, creative arts, or relaxation techniques. It may thus be possible that an emotion-focused intervention like READY-Nepal is targeting unique mechanisms that have require additional testing. In one recent meta-analysis of implementation processes for life skills programs in LMIC (a majority of which are school-based and can include emotion regulation components) [55], family participation had the greatest impact on effectiveness; our qualitative findings showed limited family involvement and discussion of skills learned, which may be an avenue of for future intervention. Other potential active ingredients in LMIC programs include problem-solving, communication skills, and self-understanding (both alone and in relation to others), which may be incorporated in future iterations of the READY-Nepal intervention. Qualitative results indicate additional modifications to consider, including use of a reward system for attendance and skills use, and additional focus on social interaction and communication skills.

A number of strengths accompanied this study. In addition, to the authors’ knowledge, ours is among the first study to examine the psychological effects of an emotion regulation intervention for adolescents in Asia. The cultural adaptation, simplification and manualization of didactic content for lay providers also enhance its dissemination and implementation potential in other resource-strained contexts in Asian settings. Its scalable nature is one key component broader global mental health agenda emphasizing delivery of culturally sensitive, effective, and appropriate care for vulnerable populations.

Limitations also exist. One is the use of a non-randomized control group, which has the potential to result in lack of baseline equivalence between the intervention and control conditions on a number of variables. Use of a non-randomized design may also result in selection bias and influence of non-random variables on study enrollment and, ultimately, study findings. Our study design study was also not powered for inference testing. The two-measurement, pre-post design also precluded any assessment of the durability of effects beyond four-weeks post-intervention. Our use of culturally modified, comparatively brief scales (with no established clinical cutoffs) further compromises the strength of any potential inferences and limits our ability to compare findings with those from other trials in LMICs. Future testing with more rigorous experimental control and improved internal validity is therefore warranted to corroborate (or nullify) current findings. Further testing (e.g., exploratory moderation analyses) is also merited, to examine profiles of individuals who benefitted from the intervention.

## 5. Conclusions

Regulating Emotions through Adapted Dialectical Behavior Skills for Youth (READY-Nepal) is a brief (8-session) intervention targeting universal prevention of mental health symptoms among disaster-exposed youth in Nepal. In a mixed-method, gender-stratified, non-randomized controlled trial of READY-Nepal, we found that the program was feasible to implement and acceptable among school-going adolescents. Qualitative results indicated student retention of anger management skills and low utilization of mindfulness skills. Despite promising feasibility and acceptability, we found no significant quantitative improvements in either primary or secondary outcomes between conditions. Future research should utilize longitudinal outcome measurement, alternatives to self-report, and rigorously develop culturally grounded models of emotion regulation to explore the utility of school-based emotion regulation interventions in Nepal and other LMICs.

## Figures and Tables

**Figure 1 ijerph-19-14497-f001:**
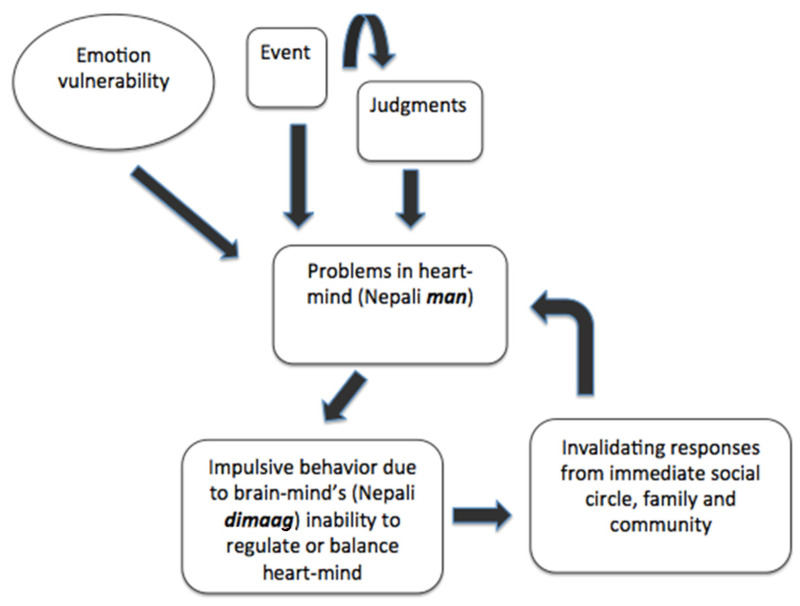
Nepali Transactional Model of Emotion Dysregulation.

**Figure 2 ijerph-19-14497-f002:**
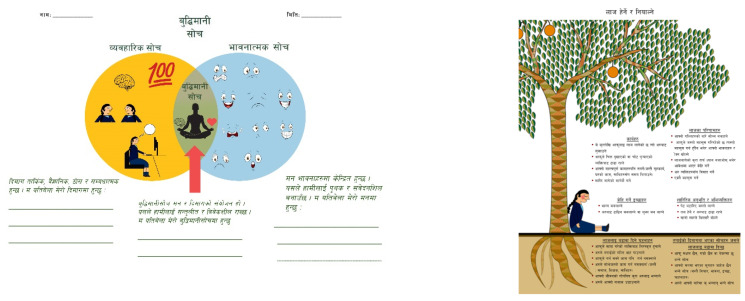
Sample handouts for READY-Nepal curriculum.

**Figure 3 ijerph-19-14497-f003:**
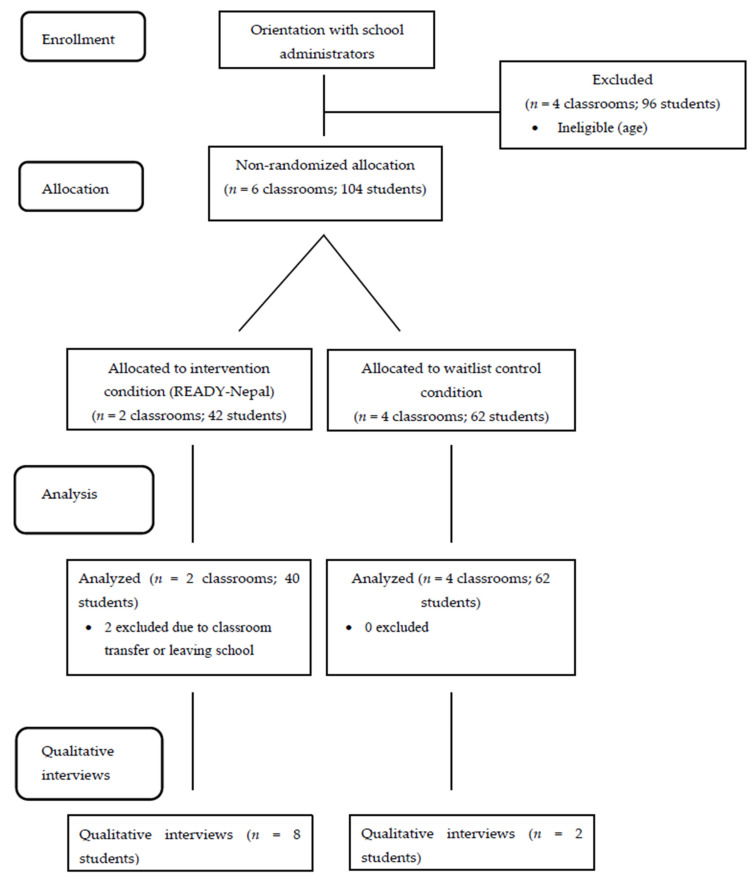
CONSORT flow diagram for overall research procedures.

**Figure 4 ijerph-19-14497-f004:**
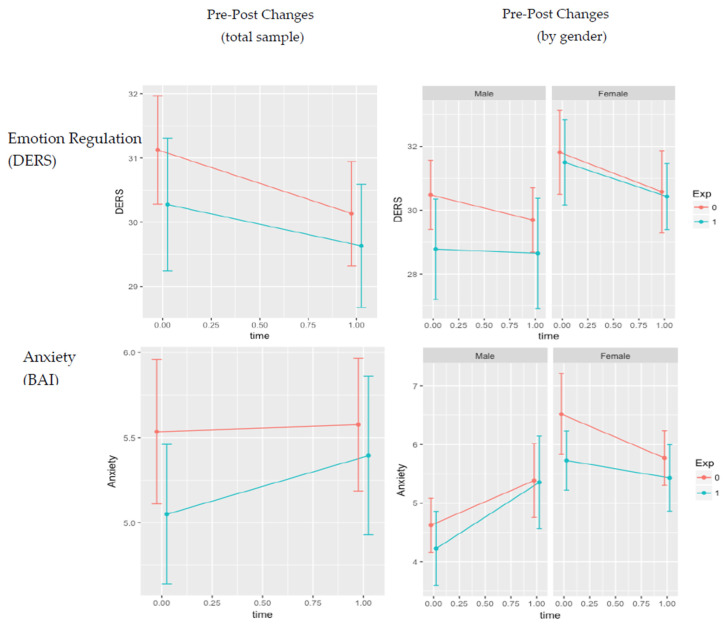
Changes in pre- and post- intervention scores by outcome measure. Note. DERS = Difficulties in Emotion Regulation Scale; DBT-WCCL = DBT Ways of Coping. Checklist; BAI = Beck Anxiety Inventory; CPSS = Child Post-Traumatic Stress Disorder Scale; CFI = Child Functioning Impairment Scale; RES = Resilience Scale.

**Table 1 ijerph-19-14497-t001:** Regulating Emotions through Adapted Dialectical Behavior Skills for Youth in Nepal (READY-Nepal) Curriculum.

Time	Topics Covered
Session One	*Diary Card* *Program Orientation* *Group Guidelines* *Transactional Model of Stress* *Mindfulness* * *Defining mindfulness*
Session Two	*Mindfulness* * *“What” skills* *“How” skills*
Session Three	*Emotion Regulation* * *Awareness of emotions*
Session Four	*Emotion Regulation* * *Changing emotions (crisis survival skills)*
Session Five	*Emotion Regulation ** *Changing emotions (opposite action)*
Session Six	*Interpersonal Effectiveness* * *Balancing priorities in relationships* *Relationship effectiveness*
Session Seven	*Interpersonal Effectiveness* * *Objective effectiveness*
Session Eight	*Program Recap & Review* *Closing*

***** Module derived from second-edition Skills Training Manual [29].

**Table 2 ijerph-19-14497-t002:** Correlation Coefficients for Baseline Variables (*N* = 102).

	(2)	(3)	(4)	(5)	(6)	(7)	(8)
(1) Emotion Regulation	0.633 **	0.564 **	0.629 **	0.358 **	0.274 **	0.399 **	0.181
(2) Anxiety	--	0.624 **	0.660 **	0.332 **	0.384 **	0.273 **	0.109
(3) Post-Traumatic Stress	--	--	0.593	0.254 **	0.273 **	0.326 **	0.182
(4) Functional Impairment	--	--	--	0.131	0.214 *	0.297 **	0.250 *
(5) Resilience	--	--	--	--	0.479 **	0.333 **	0.186
(6) Functional Coping	--	--	--	--	--	0.493 **	−0.172
(7) Dysfunctional Coping	--	--	--	--	--	--	0.194
(8) Suicidal Ideation	--	--	--	--	--	--	--

* *p* < 0.05, ** *p* < 0.01.

**Table 3 ijerph-19-14497-t003:** Baseline Characteristics by Condition (*N* = 102).

	Control (*n* = 60)	READY-Nepal (*n* = 42)	χ^2^, *t*
Age, mean (SD)	14.25 (1.40)	14.38 (1.13)	−0.50
Female, *n* (%)	28 (46.7)	23 (54.8)	0.65
Caste, *n* (%)			13.48 **
Traditionally high-caste groups (Brahman, Chhetri)	14 (23.3)	3 (7.1)	
Traditionally low-caste groups (Dalit)	3 (5.0)	1 (2.4)	
Ethnic groups	35 (58.3)	38 (90.5)	
Other	8 (13.3)	0 (0.0)	
Clinical distress			
Emotion regulation, mean (SD)	20.68 (7.60)	19.69 (7.43)	0.66
Anxiety, mean (SD)	5.45 (3.11)	5.05 (2.62)	0.69
Post-traumatic stress, mean (SD)	5.35 (3.15)	5.21 (2.75)	0.23
Functional impairment	20.87 (5.28)	18.52 (4.42)	2.36 *
Resilience	10.02 (3.75)	10.19 (4.15)	−0.22
Functional Coping	23.50 (8.91)	23.79 (8.30)	2.32
Dysfunctional Coping	3.54 (2.19)	3.41 (1.80)	−0.50
Prior two weeks’ suicidal ideation, *n* (%)	12 (20.0)	7 (16.7)	0.18

* *p* < 0.05, ** *p* < 0.01.

**Table 4 ijerph-19-14497-t004:** Univariate Tests for Time by Condition on Outcomes (*N* = 102).

	Pre-Intervention		Follow-Up				
Measure	Control *M (SD)*	READY-Nepal *M (SD)*	Control *M (SD)*	READY-Nepal *M (SD)*	*df*	*F*	Partial η^2^
Emotion Regulation	20.68 (7.60)	19.69 (7.43)	19.64 (6.32)	19.44 (5.71)			
Time					1, 92	2.08	0.022
Time*Gender					2, 92	0.22	0.002
Time*Condition					2, 92	0.78	0.008
Anxiety	5.45 (3.11)	5.05 (2.62)	5.52 (2.83)	5.38 (2.83)			
Time					1, 92	0.15	0.002
Time*Gender					2, 92	6.74 *	0.068
Time*Condition					2, 92	0.59	0.006
Trauma Distress	5.35 (3.15)	5.21 (2.75)	4.95 (2.68)	4.54 (2.27)			
Time					1, 92	3.65	0.038
Time*Gender					2, 92	1.41	0.015
Time*Condition					2, 92	0.00	0.000
Functional Impairment	20.87 (5.28)	18.52 (4.42)	18.79 (5.17)	18.05 (4.62)			
Time					1, 92	8.67 **	0.086
Time*Gender					2, 92	1.54	0.017
Time*Condition					2, 92	3.00	0.032
Resilience	10.02 (3.75)	10.19 (4.15)	9.78 (4.43)	9.82 (3.44)			
Time					1, 92	0.95	0.010
Time*Gender					2, 92	0.25	0.003
Time*Condition					2, 92	0.09	0.001
Functional Coping	23.52 (8.65)	23.55 (8.09)	22.63 (7.85)	23.49 (9.04)			
Time					1, 92	0.49	0.005
Time*Gender					2, 92	0.08	0.001
Time*Condition					2, 92	0.13	0.001
Dysfunctional Coping	3.60 (2.14)	3.45 (1.95)	3.52 (2.12)	3.25 (1.86)			
Time					1, 92	0.26	0.003
Time*Gender					2, 92	0.62	0.007
Time*Condition					2, 92	0.13	0.001

* *p* < 0.05, ** *p* < 0.01.

**Table 5 ijerph-19-14497-t005:** Changes in Past Two-Week Suicidal Ideation, by Gender (*N* = 102).

	Pre-Intervention (*n*)	Follow-Up (*n*)	Direction
Control	15	15	-
*Female*	7	10	↑
*Male*	8	5	↓
READY-Nepal	13	10	↓
*Female*	9	7	↓
*Male*	4	3	↓

**Table 6 ijerph-19-14497-t006:** Qualitative Assessment of Feasibility and Acceptability (*n* = 15).

Theme	Subtheme	Illustrative Quotations (Female Students)	Illustrative Quotations (Male Students)	Endorsement
Student Feedback on READY-Nepal Intervention				
Perceived Benefits (Intervention Group)		*“These skills are useful whenever I am angry with friends, or whenever family members shout at you, or whenever you feel alone.”* (Grade 7) *“In a group [of girls] is better…it is fun being with friends and doing things with friends.”* (Grade 7)	“*I used to feel angry before the program. Now, only sometimes I feel angry. I felt angry in a school last time. I controlled my anger and played with friends. I didn’t focus on the thing that made me angry*.” (Grade 9) *“Separate group of boys and girls is good. I always feel shame in front of girls but not in front of boys.”* (Grade 9)	○ *n* = 7 (88%)
Suggested Improvements (Intervention Group)		*“I didn’t understand anything about sadness...The [facilitators] should explain in a clearer way so it is easy. And with more sessions.”* (Grade 8) *“Writing poems, narrating stories…or having some plays would be better.”* (Grade 7) *“Prizes.. would help me stay in the sessions.”* (Grade 9)	*“We should learn more about how to help others, how to respect elders, love youngers.”* (Grade 9) *“18–20 year olds would be better.. because they can understand better and can solve hard questions easily.* (Grade 9) *“Parents… should receive the program also.”* (Grade 7)	○ *n* = 8 (100%)
Skills Use (Intervention Group)	○ Most Used ○ Least Used	*“I was very angry. I did opposite actions but that didn’t calm me much. Then I thought of calming my sense organs and taking some cold water in a steel glass and stretching my body.”* (Grade 7) *“I don’t want to follow the shame part... because whenever we fight with friends or when we fall down while walking we feel shame and I don’t want to get into the situation that makes me feel shame.”* *(Grade 8)* *“Wise mind was the least helpful…It was difficult to understand.”* (Grade 7)	*“I remember the opposite actions for anger. Now I try to control my anger.”* (Grade 9)	○ Most Used: *n* = 6 (75%) ○ Least Used: *n* = 3 (38%)
Student Perceptions (Control Group)	○ Positive Perceptions ○ Negative Perceptions ○ Neutral Perceptions	*“I just heard…that when we are angry we should go to an open space where nobody comes and sit there and we feel better…* *Whenever I feel angry, I go and stay in the open space in jungle for a while….By doing so, I don’t feel angry.”* (Grade 9)	*“My friends didn’t tell about… any activities they did.”* (Grade 7)	○ Positive Perceptions: 1 (50%) ○ Neutral Perceptions: 1 (50%)
Teacher and Caregiver Feedback on READY-Nepal Intervention				
Teacher Perceptions	○ Positive Perceptions ○ Negative Perceptions ○ Neutral Perceptions	“*I’ve noticed that these days while doing some work they…interact very openly with each other*.”	*“From my perspective, honestly, I haven’t seen any changes. I feel it is the same at it used to be.”* *“School teachers… should get more administration about who [READY-Nepal program facilitators] are, what they do, what benefit adolescents get. I don’t know if administration told all of us this or not.”*	○ Positive Perceptions: 1 (50%) ○ Negative Perceptions: 1 (50%) ○ Neutral Perceptions: 1 (50%)
Caregiver Perceptions	○ Positive Perceptions ○ Negative Perceptions ○ Neutral Perceptions	*“I’ve seen some changes in her and she’s also showing some interest in her studies within last 10–12 days… she was… talking about the singing bowl…to have a peaceful mind. So maybe this sound helped her to feel relieved.”* (Biological Mother of Grade 7 Participant) *“She said there was a program. But I don’t know about what the program is… She cooks food when I go to work, she cleans house, rooms, kitchen as before. I don’t know if she has stress in her mind.”* (Biological Mother of Class 8 Participant)	*“I haven’t seen [any changes]. He just said that program was to prevent worry and showed me the singing bell. I also didn’t ask much about the program as I was busy in my works.”* (Grandparent of Class 8 Participant)	○ Positive Perceptions: 1 (33%) ○ Neutral Perceptions: 1 (33%)

## Data Availability

Data are available from the first author upon request.

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
