# Peer review of "Feasibility and Acceptability of a School-Based Emotion Regulation Prevention Intervention (READY-Nepal) for Secondary School Students in Post-Earthquake Nepal"

_ijerph, 2022, doi:10.3390/ijerph192114497_

Round 1
Reviewer 1 Report
Dear Authors, I must first congratulate you for carrying out such a meaningful study in a good way. However, there are some issues that need to be addressed before publication in a high-impact factor journal such as IJERPH.
1. Introduction is fine. However, a detailed discussion in the 'Objective' section is needed. The research is on the cultural adaptation of DBT in the Nepali context, however, the justification for it is far from satisfactory. The researcher/s has quoted a Nepali Ethnopsychology model, however, there is no reference in it, and whether this model has been explored in the past as well. There could be many models/frameworks that have evolved at the local level, but not all of them might be sound enough. This needs to be clarified.
"The resulting intervention, Regulating Emotions through Adapted Dialectical Behaviour Skills for Youth (READY-Nepal)" is unclear as to where it has resulted. This needs to be explained. The intervention looks sound, and the components are as per the core elements of DBT.
The Aims of the research are clearly spelled out, and a sound Design has been used. However, why the researcher/s have used a non-randomized design, needs clarification. Please add sample items in all the measures used in the study. Qualitative design is also as per the standard. The demographic description is perfectly done. Quantitative data analysis is properly done, however, data was collected just two times. There is no problem with it as such, a few lines of justification would make the research more sound.
Results did not provide evidence for the effectiveness of the intervention, and the author/s explained it quite well. I would like to ask on all the measures used in the study, how the respondents have performed. Whether they scored low, average, or high. This may perhaps explain the results better.
If the authors were of this review (Fazel et al., 2014), the average length of sessions ranged from eight to 20 673 sessions, with a mean of 14 sessions, then why the author/s chose only 8 sessions? This needs to be explained.
Reviewer 2 Report
I was glad to have the opportunity to read about this cultural adaptation of DBT for students (presumptively) affected by the earthquake that struck the Kathmandu valley region in 2015. The description of the adaptation and the rationale for recruiting from schools was particularly well-articulated and offers a nice template for researchers doing similar work in LMIC countries. There was much to appreciate about this paper, and I have very few suggestions for improvement:
1) This paper struggles with the all-too common problem of convenience sampling from multiply stressed regions...We do not know the precise source of the students' traumatic stress/dysregulation (direct effects of the earthquake? poverty? compromised parenting? bullying and community violence?); and we therefore cannot be certain if the treatment is optimally dosed or targeted for the presenting problems (such as they are reported). Is there a selection bias at work here (and this would not necessarily be a bad thing), such that the students who choose to enroll in the treatment are also the most in need of the service? Can the authors speak to these issues in their Limitations section?
2) Table 6 is very difficult to read with its too closely spaced bullets and quotes. Would landscape format suit it better?
3) A very small point: the images depicted in figure 2 are in the incorrect order relative to the caption.
